# Psychosocial Wellbeing of Schoolchildren during the COVID-19 Pandemic in Berlin, Germany, June 2020 to March 2021

**DOI:** 10.3390/ijerph191610103

**Published:** 2022-08-16

**Authors:** Stefanie Theuring, Welmoed van Loon, Franziska Hommes, Norma Bethke, Marcus A. Mall, Tobias Kurth, Joachim Seybold, Frank P. Mockenhaupt

**Affiliations:** 1Institute of Tropical Medicine and International Health, Charité—Universitätsmedizin Berlin, Corporate Member of Freie Universität Berlin and Humboldt-Universität zu Berlin, 13353 Berlin, Germany; 2Medical Directorate, Charité—Universitätsmedizin Berlin, Corporate Member of Freie Universität Berlin and Humboldt-Universität zu Berlin, 10117 Berlin, Germany; 3Department of Pediatric Respiratory Medicine, Immunology and Critical Care Medicine, Charité—Universitätsmedizin Berlin, Corporate Member of Freie Universität Berlin and Humboldt-Universität zu Berlin, 13353 Berlin, Germany; 4Institute of Public Health, Charité—Universitätsmedizin Berlin, Corporate Member of Freie Universität Berlin and Humboldt-Universität zu Berlin, 10117 Berlin, Germany

**Keywords:** SARS-CoV-2, coronavirus, students, school, Berlin, GAD-7, HRQoL

## Abstract

The COVID-19 pandemic and related restrictions have affected the wellbeing of schoolchildren worldwide, but the extent and duration of specific problems are still not completely understood. We aimed to describe students’ psychosocial and behavioral parameters and associated factors during the COVID-19 pandemic in Berlin, Germany. Our longitudinal study included 384 students from 24 randomly selected Berlin primary and secondary schools, assessing psychosocial wellbeing at four time points between June 2020 and March 2021. We analyzed temporal changes in the proportions of anxiety, fear of infection, reduced health-related quality of life (HRQoL), physical activity and social contacts, as well as sociodemographic and economic factors associated with anxiety, fear of infection and HRQoL. During the observation period, the presence of anxiety symptoms increased from 26.2% (96/367) to 34.6% (62/179), and fear of infection from 28.6% (108/377) to 40.6% (73/180). The proportion of children with limited social contacts (<1/week) increased from 16.4% (61/373) to 23.5% (42/179). Low physical activity (<3 times sports/week) was consistent over time. Low HRQoL was observed among 44% (77/174) of children. Factors associated with anxiety were female sex, increasing age, secondary school attendance, lower household income, and the presence of adults with anxiety symptoms in the student´s household. Fear of infection and low HRQoL were associated with anxiety. A substantial proportion of schoolchildren experienced unfavorable psychosocial conditions during the COVID-19 pandemic in 2020/2021. Students from households with limited social and financial resilience require special attention.

## 1. Introduction

Since early 2020, the COVID-19 pandemic and subsequent response measures have caused dramatic disruptions to the daily routine of children and adolescents [1,2]. Educational institutions in 191 countries were closed throughout large stretches of 2020 and 2021 as an infection control measure, affecting an unprecedented number of 1.6 billion schoolchildren and students [3]. These school closures were largely implemented before the current scientific understanding that schools are not pandemic hotspots and that school closures should be a very last resort in pandemic response [4,5,6,7,8]. The interruption to the daily structures of children and adolescents, along with varying extents of social deprivation, inevitably led to involuntary lifestyle changes [9,10]. School closures and isolation from peers can result in reduced physical activity and increased sedentary behavior and screen exposure in children and adolescents, as well as adverse physical health outcomes regarding body composition, cardio-metabolic fitness and motor competence, etc., in addition to psychosocial problems [9,10,11,12,13]. Recent research showed that during the COVID-19 pandemic, psychological disorders in adolescents such as clinically significant generalized anxiety and depression doubled compared to pre-pandemic times [2,14]. A German study from 2020 regarding the collateral damage of the pandemic, including 800,000 minors, found a 60% increase in obesity and a 35% increase in anorexia and bulimia in tertiary care [15]. Mental health outcomes have been found to be associated with low physical activity levels and increased social isolation [16,17,18]. Socio-economically disadvantaged children deserve special concern, as social, socio-affective and physical activity stimuli might be particularly lacking in their home environment [9,10].

To date, comprehensive knowledge regarding the long-term consequences of the COVID-19 pandemic on schoolchildren is still lacking. Related findings are often derived from cross-sectional studies, and are frequently based on small sample sizes and restricted outcomes [19]. Only few studies have assessed child wellbeing and worries in the later pandemic stages in 2021. Most studies on pediatric wellbeing during the pandemic and pandemic restrictions have focused on acute rather than longer-term effects, and have usually not integrally addressed the physical, behavioral and mental aspects [19,20].

The Berlin Corona School Study (BECOSS) investigated infection and transmission dynamics, along with a broad spectrum of sociodemographic, behavioral and psychosocial aspects, over a 9 months timespan in 2020 and 2021 among schoolchildren, educational staff and connected household members in Berlin, Germany. Infection-related results from the four study time points have been published earlier [8,21,22]. In the present longitudinal study, we describe students´ psychosocial and behavioral parameters during the COVID-19 pandemic in the period between June 2020 and March 2021. We aimed to analyze changes in psychosocial and behavioral parameters over time, looking at the differences between primary and secondary school children. Moreover, we explored the combination of factors that best predicted anxiety, low HRQoL, and fear of infection.

## 2. Materials and Methods

### 2.1. Study Design, Setting and Participants

This was a longitudinal cohort study among Berlin schoolchildren based on four observation time points. We randomly selected 24 schools in Berlin by dividing the 12 city districts into three socio-economic strata [21,23]. Two districts per stratum and within these, two primary and two secondary schools were randomly chosen. Due to the limited laboratory capacities during the pandemic, participation was limited to one randomly selected class per school. Classes included grades 3–5 in primary schools (8–12-year-olds) and 9–11 in secondary schools (13–17-year-olds). All children from the selected classes were eligible for participation. Additionally, we recruited the educational staff and household members of all participants [8]; these were not considered in the current analysis except for data on anxiety among adult household members, which potentially influences anxiety symptoms in children [24]. Written informed consent was sought from both the schoolchildren and their legal guardians. The storage of study data was strictly confidential.

The four study time points spanned from June 2020 to March 2021 and were chosen to reflect the differing pandemic phases. The SARS-CoV-2 community incidence in this period in Berlin, as well as holidays and school closures are shown in Figure 1.

The first study visit (T1) in June 2020 took place eight weeks after a 6-week lockdown (including school closure), with low community incidence. The school reopening was flanked by official recommendations for infection prevention and control (IPC) measures, including reduced class sizes, absence rules for symptomatic individuals, fresh-air ventilation, hand hygiene, physical distancing, and self-isolation and testing for symptomatic students and staff. The second study visit (T2) in November 2020 coincided with the peak in the second pandemic wave in Berlin, with schools operating. At that time, facemask use was obligatory when moving in hallways, but not in class. The third study round (T3) took place in February 2021, i.e., during a public lockdown including school closure. The fourth study round (T4) was conducted at the end of March 2021, 2 to 3 weeks after schools resumed instruction with split classes at half the original size, with students attending school on alternate weeks. Then, facemask wearing became obligatory in classes, and voluntary self-testing in school, applying rapid test devices twice a week, began.

### 2.2. Data Collection and Parameters

Details of the study procedures have been described previously. Initially, 385 students participated (T1), of whom 324 continued until the last study round (T4). SARS-CoV-2 infection was detected among participating students by rt-PCR in 0.3% (T1), 2.7% (T2), 0% (T3), and 0.8% (T4) [8,21,22]. Further data were collected via a structured questionnaire in a paper-based version in the first round of study and as an online version in the following three rounds, using the Research Electronic Data Capture (REDCap) tool [25]. For primary school children, we asked parents to help their children to fill in the questionnaire.

We assessed general sociodemographic and economic family background data. Family migration background was defined as grandparents from at least one parent’s side not being born in Germany, including migrant students up to the third generation. We doubled the average monthly net income of a Berlin household with one working adult in 2021 [26] to define the “high income” category as >5000€. Further parameters were after-school leisure activities including sports and meeting friends, self-rated risk perception of obtaining SARS-CoV-2 infection, fear of infection, and general anxiety symptoms. For the latter, we used the GAD-7 questionnaire for both children and parents [27,28]. Study participants were asked about the presence of seven anxiety symptoms during the last 2 weeks. Response options included “not at all,” “several days,” “more than half the days,” and “nearly every day”, weighted from 0, 1, 2, and 3, respectively. Consequently, individual anxiety scores ranged between 0 and 21. A GAD-7 score of ≥5 was defined as presenting of anxiety symptoms. The scores for anxiety severity were 5–9 for mild, 10–14 for moderate, and 15–21 for severe [29]. To assess extent of fear of infection, we created a dichotomous category by grouping “not at all” and “a little” into overall “no or low fear of infection”, and “somewhat”, “strong” and “very strong” into overall “fear of infection”.

When interpreting behavioral aspects, we considered it insufficient for children and adolescents if physical activity, i.e., indoor or outdoor sports excluding physical education in school, was performed less than three times a week [13]. Social activity in the form of meeting one or more friends for leisure time outside of the school context was considered unfavorable if carried out less than once a week over the last 14 days. In the last test round in March 2021, we expanded the study focus to include social wellbeing and additionally measured health-related quality of life (HRQoL), applying the KIDSCREEN-10 tool for children [30]. Individual scores were transformed into T scores. Low HRQOL was defined as 0.5 SD below the mean T score of pre-pandemic reference data for children [31]. The questions targeted the preceding week at the point of data collection.

### 2.3. Data Analysis and Statistics

We report on the frequencies of categorical characteristics and median values (lower quartile, Q1, upper quartile Q3) of continuous covariates of the study participants over the four time points. We tested whether frequencies of anxiety, fear of infection, low sports activity and low overall social activity differed between the four time points using Pearson’s Chi-squared test. In univariate analysis, we compared participant characteristic, looking at anxiety (as a composite parameter assessed at any of the study time points), fear of infection (composite parameter), and low HRQoL (at T4). We computed univariate odds ratios (OR) and 95% confidence intervals (CI) for categorical variables. Additionally, we explored the combination of factors that best described anxiety (composite parameter), fear of infection (composite parameter), and low HRQoL (at T4). We fitted a binomial logistic regression model including sex, age, household income, household education, family migration background, siblings, fear of infection (composite parameter for the anxiety model, at T4 only for the other models), anxiety for at least one adult in the household (composite parameter for the anxiety model, at T4 for the other models), and physical and social activities (for the HRQoL model only, at T4). Then, we used this model for a backwards stepwise selection according to the Akaike information criterion (AIC) using the R function stepAIC from the MASS package.

We further evaluated the likelihood of variables being selected by the backward stepwise selection by using a resampling (bootstrap) technique on 1000 resampled datasets (R function boot.stepAIC from the bootStepAIC package). In short, a new dataset was simulated by resampling with replacement; the full binomial logistic regression model was fitted; the backwards stepwise selection by AIC was redone on the new, full model; among the 1000 resulting datasets, the number of times that each variable was selected by the backward selection model procedure was counted.

For descriptive statistics, missing data were excluded per variable item and not included in the denominator for frequency calculation. For the logistic regression models and bootstrap method, participants with missing data in any variable were excluded from the dataset. All analyses were carried out in R version 3.6.3 (see Appendix A).

## 3. Results

We recruited 384 students at the first time point (191 primary school students; 193 secondary school students). The numbers slightly decreased to 363 students (179; 184) at T2; 327 students (158; 169) at T3; and 324 students (156; 168) at T4. At inclusion (T1), 49.7% of students (191/384) were female, and the median age was 13.5 years (range, 8–18). At students´ households, the highest educational level was tertiary in 62.7% (89/142), monthly net income was >5.000€ in 40.6% (56/138), and 17.5% (25/143) of the students had a migration background in their family. Except for age, these parameters did not change significantly over time (median age in years [Q1, Q3]; T1, 13.5 [10.0, 15.0]; T2, 14.0 [11.0, 15.0]; T3; 15.0 [11.0, 15.0]; T4, 15.0 [11.0, 15.0]).

### 3.1. Parameters of Psychosocial Wellbeing over Time

Parameters related to psychosocial wellbeing, including fear of infection, anxiety, physical activity and social activities, and HRQoL, are shown in Table 1.

During the study period of nine months, the proportion of students expressing fear of SARS-CoV-2 infection differed (χ2 = 11.0, *p* = 0.01), increasing from 29% to 41% over time. This did not differ between primary and secondary school students at any of the time points (Figure 2).

Fear of infection expressed at any time point from T1 to T4 was reported in 49.3% (77/156) of primary school students, and 53.3% (98/168) of secondary school students. Anxiety, as assessed by the GAD-7 score, also differed for each time point (χ2 = 11.0, *p* = 0.01). Anxiety initially affected 26.2% (96/367), reached 39.0% (64/164) during the lockdown (T3), and slightly decreased thereafter. Secondary school children more frequently showed anxiety than primary school children at T2 (36/103, 35.0% [95% CI, 25.8–45.0%] versus 18/91, 19.8% [12.2–29.4%]), T3 (41/81, 50.6% [39.3–61.9%] versus 23/83, 27.7% [18.4–38.6%]), and T4 (43/95, 45.3% [35.0–55.8%] versus 19/84, 22.6% [14.2–33.0%]) (Figure 2). Anxiety at any time point during the study period was observed in 46.2% (145/314, 25.2% mild, 11.8% moderate, 9.2% severe) of all children, and in 39.5% (60/152) and 52.5% (85/162) of children attending primary and secondary school, respectively (OR, 1.69; 95% CI, 1.06–2.72). Overall, the frequency of meeting friends did not significantly differ over time (χ2 = 5.9, *p* = 0.1), whereas the frequency of physical activities did (χ2 = 9.3, *p* = 0.03) (Table 1). At any time point, social contacts in the form of meeting friends were limited to ≤2/week for most children, and physical activity levels were rather low (Figure 3). Low HRQoL, assessed at only the last study time point (T4), was observed for 44% of the children.

### 3.2. Factors Associated with Parameters of Psychosocial Wellbeing

In univariate analysis, anxiety was increased in participants with female sex, increasing age, secondary school attendance, lower household income, and when adults with anxiety symptoms were present in the child´s household (Table 2). A high fear of infection at any time point was associated with anxiety; low HrQoL (measured at T4) was associated with anxiety symptoms at the same time point.

The combination of individual variables that best described each of the outcome variables of anxiety (composite parameter), fear of infection (composite parameter), low HRQoL (at T4), and unfavorable behavior in terms of physical and social activities (at T4), along with the respective selection frequency found by the bootstrap method, can be presented as follows: Anxiety: presence of adult(s) with anxiety symptoms (86%), household income (76%), sex (57%), age (54%), and presence of siblings (54%). Low HRQoL: household income (63%), and physical activities (54%). Fear of infection: age (62%).

## 4. Discussion

Growing evidence illustrates the deleterious effects of the pandemic situation and related restrictions on child and adolescent health, including adverse influences on weight, physical activity, social contacts and mental health [2,9,10,11,12,13,15]. The frequency and magnitude of this damage may vary not only with pandemic activity and the extent of IPC measures, but also with school system capacities, socio-economic background, or the attitudes and beliefs of the affected population [9,10,32]. Our report on psychosocial wellbeing among Berlin schoolchildren in the COVID-19 pandemic between June 2020 and March 2021 represents one of the first longer-term longitudinal assessments in this age group in Germany. Essentially, our findings show an increase in fear of infection and generalized anxiety over the course of the pandemic in schoolchildren, as well as an accumulation of child anxiety in households with comparatively lower income and in households where adults show anxiety themselves. Household socioeconomic characteristics were also partially linked with children´s physical activities and social contacts.

During the nine pandemic months, which included very low community incidence as well as two pandemic waves, a lockdown and school re-openings, about half of the children were affected by a moderate to strong fear of SARS-CoV-2 infection at some point. Moreover, about half the children showed symptoms of anxiety disorder. A longitudinal study from the US equally found about 40% of adolescents to have anxiety symptoms during the COVID-19 pandemic, as compared to 25% in the pre-pandemic period in the same cohort [33]. Pre-pandemic pediatric comparison data from Germany are rare; in the BELLA study within the German National Health Interview and Examination Survey for children and adolescents in 2014, a proportion of 11% of children aged 11–17 years reported symptoms of anxiety [34]. Both fear of infection and anxiety increased in prevalence over time; a possible habituation or mitigation effect was not apparent. On the other hand, fear and anxiety did not run in parallel to the pandemic’s development: anxiety peaked in the 2021 lockdown, at low community incidence, approximately three months after the peak of the second pandemic wave. It is conceivable that shifts in awareness caused by pandemic events occur with some delay and possibly also persist for some time. This is an important finding when effectively planning countermeasures regarding psychosocial impairments in children caused by a drastic event such as a pandemic. Anxiety was more present in secondary school children than in primary school children, increasing age and among girls. Plausibly, secondary as compared to primary school children experience a less sheltered daily routine, larger tutor groups, lack of a consistent reference person, and more direct access to information, including social media, all of which might contribute to a higher burden of anxiety. Children from households with a higher income were significantly less likely to experience anxiety, implying that socioeconomic status might impact perceived stability and general trust in positive life events, especially under extraordinary circumstances. This notion is supported by findings from the BELLA study, where socioeconomic status was associated with mental health problems in children and adolescents [35]. In accordance, the German LIFE study showed a stronger decrease in HRQoL among children from medium/low compared to high socioeconomic subgroups when comparing pandemic to pre-pandemic data [10]. Almost every second child at the end of our study experienced low HRQoL. Pre-pandemic data from the BELLA study [14,36] displayed low HRQoL in only 15% of children and adolescents using the KIDSCREEN tool, while Ravens-Sieberer et al. report proportions of 40.2% during the early pandemic stage in June 2020 and 47.7% in January 2021 [14], resembling the proportion we measured in our study in March 2021.

Parental psychosocial status has been recognized as an important influencing factor on children´s psychosocial wellbeing in the COVID-19 era; parental self-efficacy, family functioning, emotional regulation and social support, were identified as particular determinants of child mental health [37]. While our study could only assess the link between parental and child anxiety, the interplay of intra-family socio-affective resources and child wellbeing, as well as the pandemic´s psychosocial toll on families urgently require further investigation [38]. Research based on cohorts with a higher proportion of families with a migration background is also needed, to assess pandemic-related inequalities and vulnerabilities in more heterogeneous populations than our study cohort.

Most children were physically exercising or meeting friends no more than twice a week. Insufficient physical activity is a major public health concern [39] and was further exacerbated during the pandemic, particularly in less socioeconomically advantaged households [9,40]. At the same time, screen time increased during lockdown according to the results of several studies [41,42], and it is plausible that the attitudes of parents with a higher work and care burden towards sedentary activities of their children were less challenging. However, as the authors of the German MoMo study argue [40], physical activity and recreational screen time do not always act as functional contradictions, and hence should be assessed and targeted separately.

Our study, which was primarily designed to assess infection dynamics in the school setting [8], has several limitations. Although compliance with repeated study procedures was overall good, the questionnaire response frequency declined over time and was particularly low for some important parameters such as socioeconomic or family migration background. Participants with missing data for any of the relevant variables in the full analysis models were excluded from the backwards selection and bootstrapping, which could have had an impact on the respective results. Due to time pressure in the study-planning phase during the early pandemic stage, we used paper-based questionnaires at T1 and online questionnaires during the other study time points. We did not record the respective duration of physical activities or the exact nature of social contacts. We lack pre-pandemic comparison data on the study group, which we accompanied through nine months of the pandemic. HRQoL was only assessed at the last study point. As information was self-reported, measurement errors were possible in our study, and a reporting bias in parents who helped younger children filling in the questionnaires cannot be excluded. Generally, our findings apply to a German setting and may not be transferrable to other settings. Major strengths include the random selection of schools across Berlin, the school-based generation of empirical data, and a long observation period.

## 5. Conclusions

The COVID-19 pandemic has profoundly affected the psychosocial well-being of Berlin schoolchildren from the beginning of the pandemic to March 2021. Our study illustrates the need to strengthen children´s resilience to adverse pandemic outcomes, with social and economic family background deserving special consideration. Ultimately, societal and political efforts to account for lacking intra-family capacities are required in order to better support children’s specific needs in such exceptional times.

## Figures and Tables

**Figure 1 ijerph-19-10103-f001:**
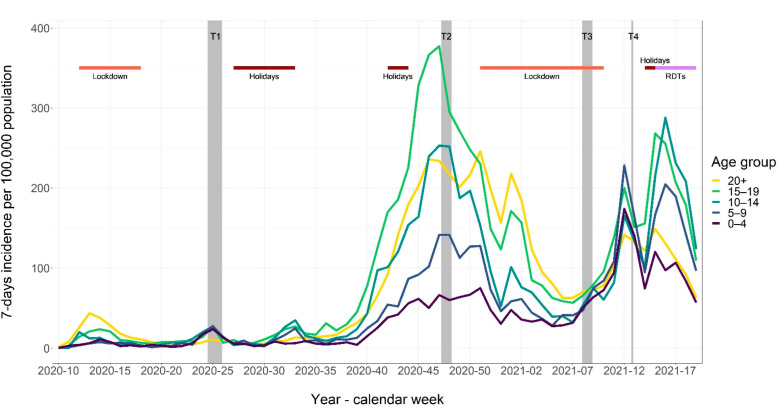
Community incidence of SARS-CoV-2 infection rates in Berlin, June 2020–March 2021, as well as observation periods (grey-shaded columns) and school holidays and closure periods (red bars).

**Figure 2 ijerph-19-10103-f002:**
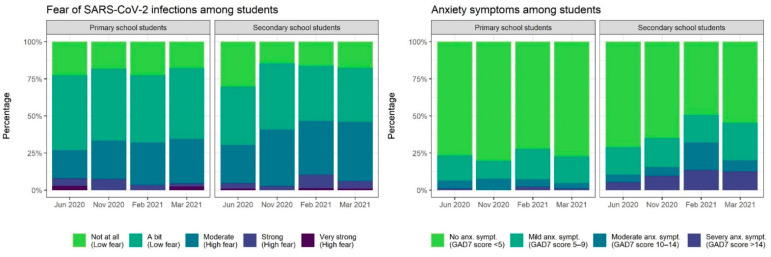
Fear of SARS-CoV-2 infection and anxiety among Berlin school children; June 2020 to March 2021.

**Figure 3 ijerph-19-10103-f003:**
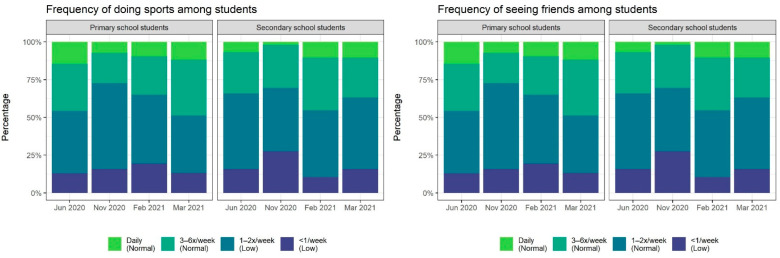
Physical and social activities among Berlin schoolchildren; June 2020 to March 2021.

**Table 1 ijerph-19-10103-t001:** Parameters related to psychosocial wellbeing among Berlin students, from June 2020 to March 2021.

	T1 (*n* = 384) (Low Incidence)	T2 (*n* = 363) (Peak 2nd Wave)	T3 (*n* = 327) (Lockdown)	T4 (*n* = 324) (Split Classes)
**Fear of SARS-CoV-2 infection**				
Low	71.4% (269/377)	62.8% (123/196)	60.6% (103/170)	59.4% (107/180)
High	28.6% (108/377)	37.2% (73/196)	39.4% (67/170)	40.6% (73/180)
Missing	7/384	167/363	157/327	144/324
**Anxiety**				
No	73.8% (271/367)	72.2% (140/194)	61.0% (100/164)	65.4% (117/179)
Yes	26.2% (96/367)	27.8% (54/194)	39.0% (64/164)	34.6% (62/179)
Missing	17/384	171/363	163/327	145/324
**Frequency of meeting friends**				
≥1 × week	83.6% (312/373)	82.8% (164/198)	77.4% (130/168)	76.5% (137/179)
<1 × week	16.4% (61/373)	17.2% (34/198)	22.6% (38/168)	23.5% (42/179)
Missing	11/384	165/363	159/327	145/324
**Frequency of physical activity**				
≥3 × week	39.9% (150/376)	29% (58/200)	40.2% (68/169)	42.5% (76/179)
<3 × week	60.1% (226/376)	71% (142/200)	59.8% (101/169)	57.5% (103/179)
Missing	8/384	163/363	158/327	145/324
**HRQoL ^a^**				
Normal or high	--	--	--	55.7% (97/174)
Low	--	--	--	44.3% (77/174)
	**T1 (** * **n** * **= 384)** **(Low Incidence)**	**T2 (** * **n** * **= 363)** **(Peak 2nd Wave)**	**T3 (** * **n** * **= 327)** **(Lockdown)**	**T4 (** * **n** * **= 324)** **(Split Classes)**
Missing				150/324

^a^ HRQoL = Health-related quality of life.

**Table 2 ijerph-19-10103-t002:** Factors associated with anxiety, fear of infection and low HRQoL among Berlin schoolchildren.

Variable		Anxiety ^a^ N = 145/314	OR ^b^	95% CI ^c^	Fear of inf. ^d^ N = 175/324	OR ^b^	95% CI ^c^	Low HRQoL ^e^ N = 77/174	OR ^b^	95% CI ^c^
**Sex**	Female	90 (57.7%)	1	-ref-	92 (57.1%)	1	-ref-	48 (49.5%)	1	-ref-
	Male	55 (34.8%)	0.39	0.24–0.63	83 (50.9%)	0.78	0.49–1.23	29 (37.7%)	0.62	0.32–1.18
**School type**	Primary	60 (39.5%)	1	-ref-	77 (49.4%)	1	-ref-	35 (43.8%)	1	-ref-
	Secondary	85 (52.5%)	1.69	1.06–2.72	98 (58.3%)	1.44	0.90–2.28	42 (44.7%)	1.04	0.55–1.98
**Age, years**	median (range)	15.0 (9.0, 18.0) ^f^	1.10	1.01–1.21	15.0 (9.0, 18.0) ^g^	1.08	0.99–1.17	15.0 (9.0, 18.0) ^h^	0.97	0.86–1.10
**HH^9^ income**	>5.000€	18 (32.1%)	1	-ref-	36 (64.3%)	1	-ref-	11 (28.9%)	1	-ref-
	≤5.000€	43 (53.8%)	2.45	1.14–5.35	48 (60.0%)	0.41	0.07–1.66	32 (47.1%)	0.80	0.19–3.57
**Highest HH^9^ education**	Tertiary	39 (43.8%)	1	-ref-	54 (60.7%)	1	-ref-	25 (36.2%)	1	-ref-
	Below tertiary	24 (47.1%)	1.14	0.54–2.41	31 (60.8%)	1.00	0.47–2.17	21 (51.2%)	1.85	0.78–4.36
**Family migration background**	No	49 (43.8%)	1	-ref-	65 (58.0%)	1	-ref-	36 (42.9%)	1	-ref-
	Yes	14 (48.3%)	1.20	0.49–2.95	21 (72.4%)	1.90	0.73–5.38	10 (37.0%)	0.78	0.29–2.08
**Siblings in HH ^i^**	No	67 (50.4%)	1	-ref-	77 (55.8%)	1	-ref-	36 (45.0%)	1	-ref-
	Yes	78 (43.3%)	0.75	0.47–1.21	97 (52.4%)	0.87	0.55–1.39	41 (43.6%)	0.95	0.50–1.80
**Fear of infection**	No	50 (35.2%)	1	-ref-	-	-	-	44 (42.3%)	1	-ref-
	Yes	95 (55.2%)	2.27	1.4–3.68	-	-	-	33 (47.1%)	1.22	0.63–2.34
**Anxiety symptoms**	No	-	-	-	77 (45.6%)	1	-ref-	31 (27.2%)	1	-ref-
	Yes	-	-	-	95 (65.5%)	2.27	1.40–3.68	46 (78.0%)	9.47	4.28–21.54
**Adults in household report anxiety symptoms ^j^**	No	15 (27.3%)	1	-ref-	30 (54.5%)	1	-ref-	18 (36.7%)	1	-ref-
	Yes	67 (53.2%)	3.03	1.45–6.49	79 (62.7%)	1.40	0.70–2.79	27 (44.3%)	1.37	0.59–3.18

^a^ Defined as GAD-7 score 5 or above, at any time point between T1 and T4; ^b^ OR = univariate odds ratio; ^c^ CI = Confidence interval; ^d^ Fear of infection, defined as “somewhat”, “strong” and “very strong”, at any time point between T1and T4; ^e^ Health-related quality of life, defined as 0.5 SD below the mean T score of pre-pandemic reference data for children, measured only at T4; ^f^ Median for group without anxiety, 12.0 (9.0, 18.0); ^g^ Median for group with low fear of infection, 12.0 (9.0, 18.0); ^h^ Median for group with normal to high HRQoL, 15.0 (10.0, 18.0); ^i^ HH = Household; ^j^ At any of the four time points.

## Data Availability

The data presented in this study are available on request from the corresponding author. The data are not publicly available due to study participants not being of age.

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
