# Peer review of "Psychosocial Wellbeing of Schoolchildren during the COVID-19 Pandemic in Berlin, Germany, June 2020 to March 2021"

_ijerph, 2022, doi:10.3390/ijerph191610103_

Round 1
Reviewer 1 Report
This is a very important and well written manuscript, shedding light on the psychosocial wellbeing of schoolchildren during the COVID-19 pandemic in a large city in Germany. Overall, the manuscript is very interesting to read and is a good contribution to the field. It details important information for policy makers and future studies as well. However, I would encourage the authors to methodologically employ a more rigorous approach for some variables in some analyses.
ll. 133 – 136 Why did you choose to dichotomize the fear of infection groups? This approach ultimately lacks power (this equals to throwing away one to two thirds of your original data and run a test on the remaining data) and makes the interpretation of results more complicated. I would be afraid that valuable information from your great dataset is lost. This also concerns the handling of GAD-7 scores. It is really interesting to see the actual distribution of scores per measurement as in Figure 2. By dichotomizing, valuable information is unfortunately lost. In Figure 2 you use all subgroups for fear of infection and anxiety symptoms (and not just the dichotomized groups) to show the development of those variable responses over time. To my mind, it would be more comprehensible if the statistical test that is used for the analysis of such time trends concerns these subgroups as well. This means dropping the dichotomizing of groups and run the appropriate test.
ll. 148 ff. The R package versions should be stated as well. R packages can change quite drastically in their development, but if you reference the actual version number, the results can be reproduced.
ll. 174 – 176 Listwise case deletion can lead to serious bias in regression models, i.e. in logistic regression as well. Do I understand you correctly, that a participant with a missing value on a variable of interest was excluded even from the bootstrapping? If this is the case, this caveat needs to be explicitly stated in the discussion or you may try alternative approaches like multiple imputation which can be run with the mice package in R (https://stefvanbuuren.name/fimd/).
Table 1. I am a bit puzzled by the numbers, maybe I read the table wrong. You stated multiple times that you recruited, e.g., 324 students at T4. But only 174 – 180 students responded to the variables in Table 1. How can there be so many missing values? Wouldn’t it be more appropriate to state that you successfully recruited around 180 students?
ll. 195 + 203 + 213 A “Chi” symbol seems to be missing in the round brackets
ll. 158 – 164 The statistical approach to determine influential variables is, to my mind, insufficient. Again, valuable information might be lost due to dichotomizing some variables. Your analysis can easily been done in R with an ordinal regression. The function “polr” from the MASS package might be of interest for you here.
ll. 333 – 334 Data and analysis scripts for such an important study should, by default, be publicly available. For many other scientific fields (e.g., physics), it is very common to share data and analysis scripts. In fields where this is usually not the case (i.e., psychology or biology) we find ourselves in serious replication crises. 60 – 70% of psychology studies’ results cannot be replicated. Open science practices, such as sharing data and scripts, are one of the most important methods by ensuring trust and believability in data and results. Since you are willing to share data upon request, I don’t see a reason to not share data right away, e.g., via the Open Science Framework. To not share data because of participants of minor age is, to my mind, a weak argument because data in Germany may only be collected after an informed consent. Somebody, probably a parent, has consented for data to be used in a scientific way. This approach constitutes good science and should therefore be no problem if the data is anonymized. This data could only include relevant variables (which are few) to simply reproduce the results you presented in your manuscript.
Author Response
ijerph-1835571: Response to Reviewer 1
Reviewer´s comment: This is a very important and well written manuscript, shedding light on the psychosocial wellbeing of schoolchildren during the COVID-19 pandemic in a large city in Germany. Overall, the manuscript is very interesting to read and is a good contribution to the field. It details important information for policy makers and future studies as well. However, I would encourage the authors to methodologically employ a more rigorous approach for some variables in some analyses.
ll. 133 – 136 Why did you choose to dichotomize the fear of infection groups? This approach ultimately lacks power (this equals to throwing away one to two thirds of your original data and run a test on the remaining data) and makes the interpretation of results more complicated. I would be afraid that valuable information from your great dataset is lost. This also concerns the handling of GAD-7 scores. It is really interesting to see the actual distribution of scores per measurement as in Figure 2. By dichotomizing, valuable information is unfortunately lost. In Figure 2 you use all subgroups for fear of infection and anxiety symptoms (and not just the dichotomized groups) to show the development of those variable responses over time. To my mind, it would be more comprehensible if the statistical test that is used for the analysis of such time trends concerns these subgroups as well. This means dropping the dichotomizing of groups and run the appropriate test.
Authors´ Response: Thank you for bringing up the dichotomization of two important response variables, and we acknowledge your concern. Indeed, power is lost by doing so and an ordinal regression of itemized groups instead of analysing dichotomized groups would have been appropriate if we analysed fear of infection and anxiety at individual time points.
However, for both variables, we combined information over the entire study period, i.e., we used the composite variables of four time points. This was done in order to describe fear of infection and anxiety “at any time point during the 9 months” and thereby present an overall mental burden from a longer-term perspective rather than the burden of specific pandemic phases (which was done in other studies). To do this, in our view, the best way is by categorizing the variables in two values, namely those below and above a threshold. On the other hand, as you correctly mentioned, we created Figure 2 to give a good insight in all the information available from our dataset. It shows the individual distribution of all fear and anxiety levels per time point and shows their time development. We do discuss these time trends also with respect to the pandemic landscape. Apart from that, we would suggest abstaining from analysing time trends of the itemized groups in function of socio-economic factors, because we believe this information does not contribute to a better understanding of the psychosocial wellbeing of children during the pandemic.
- 148 ff. The R package versions should be stated as well. R packages can change quite drastically in their development, but if you reference the actual version number, the results can be reproduced.
We agree and we added the package versions in the accompanying R Markdown script.
ll. 174 – 176 Listwise case deletion can lead to serious bias in regression models, i.e. in logistic regression as well. Do I understand you correctly, that a participant with a missing value on a variable of interest was excluded even from the bootstrapping? If this is the case, this caveat needs to be explicitly stated in the discussion or you may try alternative approaches like multiple imputation which can be run with the mice package in R (https://stefvanbuuren.name/fimd/).
Participants with a missing value on any of the variables of interest were excluded from the backwards selection analysis and from the bootstrapping. We mention this now more explicit in the discussion, ll. 316-318:
“Participants with missing data for any of the relevant variables in the full analysis models were excluded from the backwards selection and bootstrapping, which could have had an impact on the respective results.”
We chose not to employ imputation, because the underlying, multifactorial missingness pattern is unclear and might be strongly affected by variables that we don’t know (e.g., timing of individual questionnaire emails and reminders being sent out).
Nevertheless, thank you for pointing out the new version of Van Buuren’s new edition. We were still only aware of the former MICE package + literature.
Table 1. I am a bit puzzled by the numbers, maybe I read the table wrong. You stated multiple times that you recruited, e.g., 324 students at T4. But only 174 – 180 students responded to the variables in Table 1. How can there be so many missing values? Wouldn’t it be more appropriate to state that you successfully recruited around 180 students?
Thank you for bringing up this important issue. Indeed, we recruited over 300 students, meaning that we collected a biological sample from them for the primary aim of the BECOSS study, i.e., assessing SARS-CoV-2 prevalence and transmission in schools (published earlier, e.g. https://doi.org/10.2807/1560-7917). For the missing values, there might be different explanations. From T2 on, we employed an online questionnaire instead of paper-based, with a dramatic effect on the response rate. At the same time, loss to follow-up is a frequent observation in longitudinal studies. We believe that the frequent application of the questionnaire over the 9 month-period in combination with pandemic fatigues lead to a kind of questionnaire fatigue in the participants, and their willingness to fill in the form was reduced after the 2nd or 3rd time they had done so.
However, here, we report the same recruited totals as in our previous papers discussing the infection parameters in our study cohort. In our opinion, this is more coherent, given that the BECOSS study includes several sub-studies all based on the same cohort.
- 195 + 203 + 213 A “Chi” symbol seems to be missing in the round brackets.
Indeed, thanks. We changed it accordingly.
ll. 158 – 164 The statistical approach to determine influential variables is, to my mind, insufficient. Again, valuable information might be lost due to dichotomizing some variables. Your analysis can easily been done in R with an ordinal regression. The function “polr” from the MASS package might be of interest for you here.
Please see our response to the first comment above on dichotomizing the response variables. We do agree that an ordinal regression would be best to analyse single time points. However, for this work, we have combined the responses of four time points in order to get an insight of school children´s mental health status from a longer-term perspective, and believe that dichotomization by using a threshold for anxiety and fear of infection is the best solution for this.
- 333 – 334 Data and analysis scripts for such an important study should, by default, be publicly available. For many other scientific fields (e.g., physics), it is very common to share data and analysis scripts. In fields where this is usually not the case (i.e., psychology or biology) we find ourselves in serious replication crises. 60 – 70% of psychology studies’ results cannot be replicated. Open science practices, such as sharing data and scripts, are one of the most important methods by ensuring trust and believability in data and results. Since you are willing to share data upon request, I don’t see a reason to not share data right away, e.g., via the Open Science Framework. To not share data because of participants of minor age is, to my mind, a weak argument because data in Germany may only be collected after an informed consent. Somebody, probably a parent, has consented for data to be used in a scientific way. This approach constitutes good science and should therefore be no problem if the data is anonymized. This data could only include relevant variables (which are few) to simply reproduce the results you presented in your manuscript.
We agree that reproducibility of analyses is important and are happy to publish the dataset and accompanying analysis script with the paper.
Reviewer 2 Report
Dear Authors!
Thank you for the opportunity to review this manuscript. It was thematically relevant and important, an logically written. It contained important information regarding long-term consequences of the COVID-19 pandemic on school children. The gap of the research has been identified. The research question was well-defined and the results provide knowledge of long-term consequences. Conclusions are justified and supported by the results.
The study is correctly designed and technically sound, but the data collection could be presented more in detail to gain insight and assessment of reliability. For example
- more information of why HRQoLa was used only on T4?
- this was a longitudinal cohort study among Berlin school children based on four observation time points, it would be good to know on what basis these points were chosen
- and also on what basis one class was selected per school.
In the evaluation of reliability, it would be good to elaborate why was there paper-based version and then online version in other three rounds. And the help of parents for primary school children is understandable, but does it have any effect on reliability?
The ethical concerns relate to the description of how participants in the study were recruited and how consent was sought. It would be good to mention of the confidential storage of the study data.
It would be good to know more about were there any inclusion criteria for schoolchildren.
In limitations it might be good to mention, since the study was conducted under German conditions, that its results cannot be generalized as such in other settings.
It would be good to know more about why the questionnaire response frequency declined over time and was particularly low for some important parameters such as socioeconomic or family migration background?
Though the discussion is versatile and the conclusions are interesting, it might benefit from a broader perspective. In discussion it could be relevant to reflect more about the researchers' own views based on the study, how children's well-being could be better protected in times like a pandemic.
It would also be interesting to know what further research ideas the authors had on the subject in addition to this pandemic´s psychosocial toll on families. What would be essential in the authors' opinion, for example when you think about socioeconomic or family migration background (as Family migration background was defined as grandparents from at least one parent side not born in Germany, capturing students belonging up to third migrant generation). Maybe this article could be helpful https://www.mdpi.com/1715478
This was an important observation: fear and anxiety did not run in parallel to the pandemic development. Could there be something to learn from this?
Specific comments:
Line 139 brackets missing in ref. 13?
Line 263 reference 34 place after full stop misplaced.
Important issues has been found in the manuscript, and it could be supplemented by examining the issues that came up in this review.
Hopefully these comments are helpful to you, wishing all the best!
Author Response
Reviewer 2 comment: The study is correctly designed and technically sound, but the data collection could be presented more in detail to gain insight and assessment of reliability. For example
- more information of why HRQoLa was used only on T4?
Authors´ response: The study was initially designed as to measure SARS-COV-2 transmission and infection dynamics in schoolchildren. We published respective results earlier, for example in: Theuring S, Thielecke M, van Loon W, et al: SARS-CoV-2 infection and transmission in school settings during the second COVID-19 wave: a cross-sectional study, Berlin, Germany, November 2020. Euro Surveill. 2021: 26(34). https://doi.org/10.2807/1560-7917.
During the data collection for this assessment of infection dynamics, it became clear that not many children were actually infected, especially after T3 when a lockdown had been in place. Therefore, in the later phase of data collection, we expanded the focus of the study on mental health and social wellbeing aspects of the pandemic, and HRQoL questions were included at T4. We included a phrase in methods section: “In the last test round in March 2021, we expanded the study focus towards social wellbeing and additionally measured HRQoL” (lines 144f).
- this was a longitudinal cohort study among Berlin school children based on four observation time points, it would be good to know on what basis these points were chosen
The four test rounds time points were chosen in a content-related approach, according to the varying pandemic situations and phases. After T1, there was a low incidence phase were a test round did not seem reasonable, so we scheduled T2 in the beginning of the second COVID-19 wave in Germany. T3 was chosen because there was a lockdown taking place, as a contrast to school presence, and T4 was chosen because it was right after the lockdown, so a potential impact of lockdown on transmission dynamics and mental health was assumed to be visible. We added a short explanation on this rationale in methods section (”The four study time points spanned June 2020 to March 2021 and were chosen in order to reflect differing pandemic phases.” - line 90f).
- and also on what basis one class was selected per school.
Due to limited lab capacities especially in the onset phase of the pandemic, we could not include more than one class per school, in order to not obstruct important health system infrastructure. This was a logistic decision. We included the following phrase: “Due to limited laboratory capacities during the pandemic, participation was limited to one randomly selected class per school” (Line 81ff)
In the evaluation of reliability, it would be good to elaborate why was there paper-based version and then online version in other three rounds. And the help of parents for primary school children is understandable, but does it have any effect on reliability?
The paper-based questionnaire at T1 was basically a logistic decision due to limited planning time in the beginning of the study. Research taking place during an emerging pandemic oftentimes lacks sufficient lead time for study preparation and requires fast and flexible research planning. This is the reason why we used the online version only from T2 onwards. For parental response bias in younger children, it is true we could not evaluate their response reliability. We added a phrase on this behalf in the limitations section. “Due to time pressure in the study planning phase during the early pandemic stage, we used paper-based questionnaires at T1 and online questionnaires during the other study time points.” (Lines 319f), “…and a reporting bias in parents who helped younger children filling in the questionnaires cannot be excluded.” (Lines 325f)
The ethical concerns relate to the description of how participants in the study were recruited and how consent was sought. It would be good to mention of the confidential storage of the study data.
We included the following phrase in methods: “Written informed consent was sought from both the school children and their legal guardians. Storage of study data was strictly confidential.” (Line 88ff)
It would be good to know more about were there any inclusion criteria for schoolchildren.
There were no exclusion criteria within the selected classes if the participants gave their informed consent. We included this in line 84f: “All children from the selected classes were eligible for participation.”
In limitations it might be good to mention, since the study was conducted under German conditions, that its results cannot be generalized as such in other settings.
This was added to the limitations section: “Generally, our findings apply to a German setting and may not be transferrable to other settings.” (Line 326f)
It would be good to know more about why the questionnaire response frequency declined over time and was particularly low for some important parameters such as socioeconomic or family migration background?
We believe that the frequent application of the questionnaire over the 9 month-period in combination with pandemic fatigues lead to a kind of questionnaire fatigue in the participants, and their willingness to fill in the form was reduced after the 2nd or 3rd time they had done so. This is a frequent observation in longitudinal studies. For socioeconomic parameters, we believe that participants might have felt this information was too private, or even unimportant for the study, many of them being primarily interested in their infection status. However, since this is very speculative, we do not think it is reasonable to comment on it in the text apart from mentioning it in the limitations (line 317f).
Though the discussion is versatile and the conclusions are interesting, it might benefit from a broader perspective. In discussion it could be relevant to reflect more about the researchers' own views based on the study, how children's well-being could be better protected in times like a pandemic.
Thank you very much for this suggestion. We actually had discussed this beforehand, and had then decided to leave out our own views, since the topic is so controversially discussed and highly sensitive in Germany. It therefore appeared a better approach to just deliver our findings and let them speak for themselves. However, the last sentence of our conclusion gives an insight into the researchers´ view on the sometimes insufficient support of school children´s wellbeing in Germany (because a lot of responsibility is left on the shoulders of individual families, regardless of their capacities), and the need for societal structures to compensate for intra-family gaps. It reads: “Ultimately, societal and political efforts to account for lacking intra-family capacities are required in order to better support children regarding their specific needs in exceptional times like this.”
It would also be interesting to know what further research ideas the authors had on the subject in addition to this pandemic´s psychosocial toll on families. What would be essential in the authors' opinion, for example when you think about socioeconomic or family migration background (as Family migration background was defined as grandparents from at least one parent side not born in Germany, capturing students belonging up to third migrant generation). Maybe this article could be helpful https://www.mdpi.com/1715478
Since in our data analysis, the family migration background did not play a role as an associated factor with anxiety of HRQoL, we refrained from further interpreting or commenting on this aspect in the discussion. However, your comment is pointing to a very justified need for research, and we included the following phrase in the discussion: “Also, research based on cohorts with a higher proportion of families with migration background is strongly needed, in order to assess pandemic-related inequalities and vulnerabilities in more heterogeneous populations than our study cohort.” (Line 300ff)
This was an important observation: fear and anxiety did not run in parallel to the pandemic development. Could there be something to learn from this?
Indeed, this result indicates that health policy makers should consider this time gap when offering psychosocial health services. We added the following phrase: “This is an important finding when it comes to effectively planning countermeasures for children´s psychosocial impairment through a drastic event like a pandemic.” (Line 276f)
Specific comments:
Line 139 brackets missing in ref. 13?
Added.
Line 263 reference 34 place after full stop misplaced.
Fullstop relocated.
Important issues has been found in the manuscript, and it could be supplemented by examining the issues that came up in this review.
We included all the comments from your very helpful revision. We think the manuscript has largely benefitted from these comments. Thank you very much for giving us the opportunity to revise!